# Susceptibility of Mediterranean Buffalo (*Bubalus bubalis*) following Experimental Infection with Lumpy Skin Disease Virus

**DOI:** 10.3390/v16030466

**Published:** 2024-03-19

**Authors:** Elisabetta Di Felice, Chiara Pinoni, Emanuela Rossi, Giorgia Amatori, Elisa Mancuso, Federica Iapaolo, Angela Taraschi, Giovanni Di Teodoro, Guido Di Donato, Gaetano Federico Ronchi, Maria Teresa Mercante, Mauro Di Ventura, Daniela Morelli, Federica Monaco

**Affiliations:** 1Istituto Zooprofilattico Sperimentale dell’Abruzzo e del Molise G. Caporale, 64100 Teramo, Italy; c.pinoni@izs.it (C.P.); e.rossi@izs.it (E.R.); gamatori@unite.it (G.A.); elisa.mancuso@iss.it (E.M.); f.iapaolo@izs.it (F.I.); a.taraschi@izs.it (A.T.); g.diteodoro@izs.it (G.D.T.); f.ronchi@izs.it (G.F.R.); t.mercante@izs.it (M.T.M.); m.diventura@izs.it (M.D.V.); d.morelli@izs.it (D.M.); f.monaco@izs.it (F.M.); 2Servizio Veterinario Igiene degli Allevamenti e Produzioni Zootecniche, ASL2 Lanciano Vasto Chieti, 66054 Vasto, Italy; 3Istituto Superiore di Sanità, 00161 Rome, Italy

**Keywords:** lumpy skin disease, buffalo, ELISA, real-time PCR, VN test

## Abstract

Lumpy skin disease (LSD) is a viral disease of cattle and water buffalo characterized by cutaneous nodules, biphasic fever, and lymphadenitis. LSD is endemic in Africa and the Middle East but has spread to different Asian countries in recent years. The disease is well characterized in cattle while little is known about the disease in buffaloes in which no experimental studies have been conducted. Six buffaloes and two cattle were inoculated with an Albanian LSD virus (LSDV) field strain and clinically monitored for 42 days. Only two buffaloes showed fever, skin nodules, and lymphadenitis. All samples collected (blood, swabs, biopsies, and organs) were tested in real-time PCR and were negative. Between day 39 and day 42 after inoculation, anti-LSDV antibodies were detected in three buffaloes by ELISA, but all sera were negative by virus neutralization test (VNT). Cattle showed severe clinical signs, viremia, virus shedding proven by positive real-time PCR results, and seroconversion confirmed by both ELISA and VNT. Clinical findings suggest that susceptibility in buffaloes is limited compared to in cattle once experimentally infected with LSDV. Virological results support the hypothesis of buffalo resistance to LSD and its role as an accidental non-adapted host. This study highlights that the sensitivity of ELISA and VNT may differ between animal species and further studies are needed to investigate the epidemiological role of water buffalo.

## 1. Introduction

Lumpy skin disease (LSD) is a viral disease of cattle and water buffalo characterized by multifocal cutaneous nodules, biphasic fever, oral and nasal discharge, and lymphadenitis. Clinical infections range from mild subclinical to acute and are often influenced by age, breed, immune status, and production period of animals [1]. LSD is associated with moderate-to-high morbidity and low mortality. Despite that, the disease has a high socio-economic impact, resulting in a decrease in milk production, decreased weight gain, permanent damage to hides, reduced reproduction due to increased infertility and abortion, costs for surveillance activity, and serious trade restrictions [2,3,4]. For these reasons, the Animal Health Law (Regulation (EU) 2016/429) classified LSD in the list of diseases belonging to Category A. LSDV is subject to early notification and reporting throughout the European Union Countries, and its prevention and control measures are governed by the Regulation (EU) 2016/429 and supplemented by Delegated Regulation 2020/687 and Implementing Regulation 2021/1070.

The disease is caused by the lumpy skin disease virus (LSDV), which belongs to the family *Poxviridae*, genus *Capripoxvirus*, together with sheeppox virus (SPPV) and goatpox virus (GTPV). Transmission of LSDV occurs mainly mechanically via blood-feeding insects and ticks. In addition, direct and indirect transmission have been reported via common use of feeders or drinking troughs by infected cattle [5] as well as seminal fluid [6]. LSD diagnosis is primarily based on the clinical signs, and subsequently confirmed by the molecular analysis of lesion crusts or biopsies of the nodules [7,8,9]. However, LSDV infection is not always apparent, and mild and subclinical disease occurs when up to 50% of animals remain uninfected or subclinically infected, which is also confirmed by cattle experimentally infected [10,11]. The incubation period in field conditions varies from 2 to 4 weeks, and from 4 to 14 days in experimental disease [12,13]. The first skin lesions appear at the inoculation site after 4–20 days. In the acute form, animals develop a biphasic febrile reaction that may exceed 41 °C and last from 4 to 14 days. This is accompanied by depression, reluctance to move, inappetence, sialorrhea, nasal discharge from mucous to purulent, and tearing. Lymph nodes are enlarged, especially prescapular and precrural [12,13,14].

LSD is endemic in most of the African continent and it has recently spread throughout the Middle East, including Turkey. In 2015, the first report of LSD in Europe came from Greece, with more than 100 outbreaks reported [15]. In 2016, cases were also reported in Bulgaria, Serbia, the Former Yugoslav Republic of Macedonia and Albania [16]. Thanks to mass vaccination campaigns with homologous LSD vaccines in the infected countries of south-eastern Europe as well as in neighboring countries (Bosnia and Herzegovina, Croatia), the spread of the disease was contained, and no LSD cases have been reported since 2017. LSD outbreaks were reported in Armenia and Russia in 2015, Georgia and Kazakhstan in 2016, and later in 2019 and 2020 in Israel, Saudi Arabia, and Syria. Between July and August 2019, LSDV was introduced in Asia affecting Bangladesh, China, and India, whereas it was reported in Bhutan, Hong Kong, Myanmar, Nepal, Sri Lanka, Taiwan, and Vietnam in 2020 [17]. To date, no LSDV outbreaks have been reported in Italy.

The Italian Mediterranean buffalo is an Italian breed of water buffalo (*Bubalus bubalis*) whose number of breeding and animals are constantly increasing, making it the country with the largest number of buffaloes reared in the EU. Due to the triple aptitude of this species for milk, meat, and work, its population increased by 50% in the last ten years, representing an important economic reality with increasing potential. In particular, it is mainly raised for the production of milk, PDO mozzarella (EEC 2081/92), and ricotta cheese in the central and southern regions of Italy, such as Campania, Lazio, and Molise. Recently the demand for buffalo meat has also increased thanks to the growing appreciation of its organoleptic properties.

Under field conditions, LSDV infection in water buffalo is a controversial matter. Some studies described the isolation of LSDV from skin lesions in buffalo [18,19]. Previously, Davies [20] reported that African buffalo (*Syncerus caffer*) and Asian water buffalo (*Bubalus bubalis*) did not show lesions in the field during LSD outbreaks though they had seroconverted. A recent field study reported that blood and skin biopsy samples collected from buffaloes in outbreaks in Egypt were negative for the presence of LSDV [9].

To date, although some descriptive articles on buffaloes confirm their sensitivity to LSD [9,21,22], there is only a field study on buffaloes naturally infected by LSDV [23] and no experimental trials are reported yet. The susceptibility of water buffalo to LSDV and its role in spreading the disease are unclear, so further studies are needed to fill these gaps.

This study aimed to determine the susceptibility of buffaloes to LSDV infection and to describe the clinical, virological, and serological responses of water buffaloes following the LSDV infection.

## 2. Materials and Methods

### 2.1. Animals

A group of eight Mediterranean buffaloes (*Bubalus bubalis*), five males and three females, and two bovines were used as a positive control. All animals were between seven and eleven months old. The animals were consecutively numbered from 1 to 10 and maintained in the high-containment animal facilities (an insect-proof establishment) of the Instituto Zooprofilattico Sperimentale G. Caporale, Teramo, Italy, and housed with a 12-hourly light–dark cycle, temperature between 10 °C and 25 °C, and relative humidity between 40% and 70%. Animals were fed concentrated rations twice daily and given ad libitum access to hay and water. Environmental enrichment was provided, including rubber toys and a hollow ball stuffed with hay.

The animals, sourced from a commercial herd, were confirmed as negative for Bovine Viral Diarrhea Virus, Parainfluenza type 3 virus, Bovine Adenovirus, Bovine herpesvirus-4, Bovine herpesvirus-1, Bluetongue, *Chlamydia psittaci* and *Coxiella burnetii* prior to study commencement.

To detect any possible dipteran presence, indoor blood-feeding insect UV light traps and sticky traps were mounted at regular intervals on the walls of the high-containment animal facilities.

The respective experimental protocols were reviewed by the state ethics commission (OPBA) and approved by Italian Ministry of Health. The experimental procedures were conducted according to the Law decree n. 26, art. 31, 2014, and they were approved by the Minister of Health (n. 722 of 13 July 2020).

### 2.2. Experimental Infection

LSDV inoculation stock was obtained at “Istituto Zooprofilattico Sperimentaledell’Abruzzo e del Molise G. Caporale”. LSDV field strain named 7416/5 was isolated from a symptomatic calf skin nodule, as described by Babiuk et al. [7]. The sample was collected during an outbreak that occurred in Albania in 2017 and was kindly provided by Dr. Ledi Pite and Dr. Liljana Cara working at the Food Safety and Veterinary Institute (FSVI) in Tirana. Virus stock was obtained by five consecutive passages on MDBK (ATCC-CCL-22) cells, a continuous cell line susceptible to LSD virus infection. Cells were incubated at 37 °C in an atmosphere of 5% CO_2_ and observed daily with an inverted microscope (20–40× Leica DFC425 C, Leica Microsystem Ltd., Buccinasco, MI, Italy) to evaluate the presence of virus–specific cytopathic effect (CPE). When CPE was complete, cells were frozen at −80 °C degrees and thawed three times and suspension was harvested. After centrifugation at 300× *g* for 30 min at 4 °C, the supernatant was collected, distributed into aliquots, and stored at −80 °C until use. One aliquot was verified for sterility for bacteria, fungi and mycoplasma. Another aliquot was used to determine the viral titer by VNT, and for virus identification by qPCR.

Six of the eight buffaloes (no. 3 to 8) were randomly assigned to the inoculated group, the remaining two buffaloes (no. 1–2) were used as the control group, while the two cattle (no. 9–10) were used as positive control. The six buffaloes, from number 3 to 8, were inoculated intravenously into the jugular vein with 5 mL of LSDV field strain suspension with titer of 10^5.8^ TCID_50_/mL, and with 1 mL injected intradermally in 2 sites on each side of the neck (0.25 mL in each site). The same procedure and the same LSDV suspension were used to inoculate two calves used as positive control. The remaining two buffaloes were mock-inoculated with the same amount of supernatant of LSDV-negative cell culture using the same procedure.

### 2.3. Clinical Observation

Buffaloes and calves were daily examined for clinical signs, in particular fever, anorexia, depression, lesions, including cutaneous nodules, and lymphadenopathy during the entire trial (42 days). Body temperature was registered each day and scored as fever if ≥39.5 °C for at least two consecutive days. Other observations were collected (Table 1) and used to calculate a cumulative clinical score [11,24].

### 2.4. Samples

For molecular analysis, EDTA blood samples were collected from all animals from day 2 to day 22 post-inoculation (p.i.), whereas oral, nasal, and ocular swabs were collected daily from day 2 to day 14 p.i. and then every 3–4 days until the end of the experiment. Serum samples were collected every 3–4 days from day 3 p.i. until the end of this study for serological analysis.

Skin biopsies were carried out on three buffaloes (no. 3, 6, and 8) and bovine no.10 at different times. On day 13 p.i. a skin biopsy was collected from buffalo no. 6 from a nodule at the inoculation site. On buffalo no. 3, six biopsies were taken on days 16, 22, 23, and 25 p.i. from nodules that developed in different parts of the body (neck, thorax, dewlap, inoculum site, and intermandibolar zone). On buffalo no. 8, skin biopsies were collected from one nodule at inoculation site on day 16 p.i. and another one at the tip of the right shoulder on day 30 p.i. On day 7 p.i., skin biopsy was collected from a nodule on the neck from animal no. 10. Hair was removed from the biopsy sites with electric clippers and cleaned with skin wipes containing 2% chlorhexidine in 70% alcohol (Clinell, GAMA Healthcare, Hemel Hempstead, UK); 2.5 mL of lignocaine (Lidocaine Hydrochloride injection 2%, Hameln Pharmaceuticals, Hameln, Germany) was injected subcutaneously, and after 10 min a 0.8 cm punch biopsy was taken using a disposable biopsy punch (Integra Miltex, Princeton, NJ, USA). Half of the biopsy tissue was placed into 10% sterile buffered formalin (Merck, Darmstadt, Germany) for a minimum of 48 h. The remaining tissue was tested in real-time PCR [8]. Finally, animals were sedated with guaifenesin 80 mg/kg, rompum 50 mg/kg (xylazine hydrochloride 23.32 mg/mL, Bayer AG 51,368 Leverkusen, Germany), were anesthetized with pentothal sodium 7–13 mg/kg (tiopentalesodico MSD Animal Health S.r.l.) and then euthanized using Tanax T-61 (Mebezonium iodide 50.00 mg/mL, Embutramide 200.00 mg/mL, Tetracaine 5.00 mg/mL, MSD Animal Health S.r.l.). A panel of organs and tissues (spleen, liver, kidney, tonsils, skin, lymph nodes, heart, rumen, abomasum, ileum, testicles, ovaries, nasal mucosa, and tongue) was collected during necropsy and analyzed using the pan-capripox real-time PCR [8].

### 2.5. Serological Examination

Serological analysis was performed using the commercially available ID Screen Capripox Double Antigen ELISA (ID.vet, Montepellier, France) and VNT. The ID Screen Capripox ELISA was performed according to the manufacturer’s instructions. Samples with an S/P% ratio of ≥30% were considered positive.

For the detection of neutralizing antibodies, an LSDV-specific VNT based on a modified protocol of WOAH was performed [25,26]. For this purpose, test serum samples were inactivated at 56 °C for 30 min, and log2 dilution series in serum-free minimal essential medium from 1:5 to 1:640 were prepared in duplicate using a 96-well format (50 µL of each serum dilution/well), in order to be titrated against a constant titer of 100 TCID_50_ in 50 µL of LSDV Neethling strain. LSDV Neethling strain used for VNT is an attenuated vaccine strain obtained from OBP (Onderstepoort Biological Products). The virus was amplified using a confluent monolayer of MDBK (ATCC-CCL-22) cells at 37 °C until 90–100% CPE was observed. The harvested virus was titrated and stored at −80 °C until required. Microtiter plates with serum-virus suspension were incubated for 2 h at 37 °C in 5% CO_2_ and after incubation 100 μL of MDBK (ATCC-CCL-22) a concentration of 10^5^ cell/mL was added to each well on 96-well plate to obtain a final concentration of 10^4^ cell/well in the SN test. Plates were observed daily using an inverted microscope (20–40× Leica DFC425 C, Leica Microsystem Ltd.) to evaluate the presence of virus–specific CPE, and after 4 days at 37 °C in 5% CO_2_ the titer was determined. Wells were scored as positive for neutralization of the virus if 100% of the cell monolayer is intact. The highest dilution of serum resulting in complete neutralization of virus (no CPE) in half of the test wells is the 50% end-point titer of that serum. A titer of 1:10 or greater was considered to be positive.

### 2.6. DNA Extraction and Molecular Analysis

Organ samples and skin biopsies were homogenized in PBS plus antibiotics (10^6^ IU/L penicillin, 10 g/L streptomycin, 5 × 10^6^ IU/L nystatin, and 125 mg/L gentamicin, IZSAM) using a Tissue Lyser II homogenizer (QIAGEN, Hilden, Germany). Nasal, oral, ocular, and rectal swabs were frozen at −80 °C and thawed three times before being tested. DNA was extracted from homogenized organs and biopsy samples, EDTA blood and swab samples using BioSprint 96 One-For-All Vet Kit (Indical Bioscience, Leipzig, Germany) following the manufacturer’s instruction. Subsequently, all samples were tested using a pan-capripox real-time qPCR [8].

## 3. Results

### 3.1. Clinical Observation

Six water buffaloes and two cattle were inoculated with an LSDV field strain while two buffaloes were inoculated with a placebo. All the animals were monitored for clinical signs (Table 2) and viremia over 42 days. Two mock inoculated animals did not show any clinical signs. Instead, buffaloes no. 4, 5, and 6 had enlarged prescapular and prefemoral lymph nodes from day 5 to day 8 p.i., but they did not develop any other clinical sign apart from nodules at inoculation sites. Buffaloes no. 3 and 8 developed fever on day 14 and day 27 p.i. (Figure 1), respectively, associated with a decrease in food intake, and generalized lymphadenomegaly. After 24 h from inoculation, small nodules appeared on both sides of their neck and spread to the whole body (neck, legs, back, and flanks) on the following days (Figure 2A,B). Buffalo no. 3 developed well-circumscribed cutaneous nodules which reduced in size and disappeared from day 30 p.i. onward. Nodules in buffalo no. 8 were well-circumscribed, and some of them started to reduce in size by day 37 p.i.

In the challenged control group, typical clinical signs of LSD were observed from day 4 p.i.: calves showed pyrexia, enlargement of superficial lymph nodes, appearance of skin nodules at inoculation site and inappetence (Table 2). Calve no. 9 developed fever on day 7, while calve no. 10 had a fever from day 4 (39.7 °C) to day 8 p.i. (39.5 °C), and then increased again until day 10 p.i. (40.4 °C) when the animal was humanely euthanized due to severe clinical signs and to avoid unnecessary suffering.

Clinical score was recorded daily after virus inoculation to measure the severity of LSDV infection (Figure 3). From the day of inoculation, all animals started with score 0. The difference in clinical scores between the ones with clinical signs and the others was clear from day 14 to day 30 p.i., when the score increased in buffaloes no. 3 and 8 up to 10 and then decreased. The other buffaloes had a clinical score ranging between 0 and 5, while cattle used as the positive control group had a clinical score between 9 and 14 (from day 6 to day 9 p.i. and then started to decrease).

### 3.2. Serological Analysis

Serological analyses were conducted using ELISA and the SN test to assess the seroconversion. On the day of the inoculation (D0), all animals were seronegative. Buffaloes inoculated with the placebo remained serologically negative throughout the 42 days of the experiment. The two buffaloes with characteristic clinical signs were positive in ELISA on day 39 (no. 3, S/P 38%, 25 days after onset of fever) and on day 42 p.i. (no. 8, S/P 30%, 15 days after onset of fever), remaining positive until the end of this study. Buffalo no. 6 seroconverted on day 42 p.i. (S/P 32%) (Figure 4). During the whole study, the other inoculated buffaloes did not show any serological response and none of the infected buffaloes developed neutralizing antibodies against LSDV. Calf no. 9 from the positive control group seroconverted from day 31 p.i., and remained ELISA positive until the end of this study. The same calf developed neutralizing antibodies from day 14 (1:80) till the end of the trial. Neutralization titers peaked on day 22 p.i. (1:320) and remained strongly positive until the time of euthanasia, confirming the development of neutralizing antibodies against LSDV.

### 3.3. Molecular Analysis

No viremia was detectable in any buffalo at any time during the trial, even in febrile animals. Viral DNA was not found in skin biopsies, swabs, and organs of the challenged animals (Table 3). On the contrary, viral DNA was found in the blood sample of cattle no. 9 from day 9 to day 14 p.i. with a Ct value between 28.54 and 38.4, while viremia was detected from day 5 (Ct 35.57) to day 9 p.i. (Ct 32.97) in cattle no. 10. Real-time PCR was performed on nasal, oral, and conjunctival swabs at different times (Table 4). On day 10 after infection, calf no. 10 was euthanized and all organs and skin biopsy collected were positive by real-time PCR. After 42 days, viral DNA was found in prescapular lymph nodes of calf no. 9 (Table 3).

## 4. Discussion

There is little evidence of the susceptibility of water buffaloes to LSDV infection as well as their epidemiological role during an outbreak. In order to fill these gaps, six Mediterranean buffaloes were experimentally inoculated with LSDV isolated from skin lesions collected from Albanian cattle in 2016 to monitor the clinical, serological, and virological response evoked by the infection. The intravenous inoculation route was used because is proven to be the most effective way to produce severe generalized disease as previously described in an experimental study in cattle [4,13]. In addition, intradermal inoculation was selected to reproduce the natural route of infection.

The susceptibility of buffalo to LSDV infection is controversial. Ahmed and Dessouki [27] reported severe clinical signs in cattle infected by LSDV while water buffaloes from the same affected area appeared clinically healthy. In recent years, an observational study conducted on buffaloes during an outbreak in Egypt confirmed their low susceptibility to this virus [19]. Another paper, instead, reported evidence of natural infection with LSDV in Egyptian buffaloes [28], where animals showed typical lesions of the disease. Researchers hypothesized that the low susceptibility of this species to the disease is due to its thick skin, which would prevent the mouthparts of blood-sucking insects such as mosquitoes, flies, and ticks from easily pass through it, thus decreasing the transmission rate [23,29]. Inoculation of six buffaloes with a field strain of LSDV resulted in the development of mild generalized disease with the appearance of small skin nodules resembling the “Neethling disease” often observed after the use of attenuated vaccines [30]. However, it is important to note that the same inoculum caused a severe clinical response when used to infect cattle [27] showing morbidity consistent with previous studies [12,13] where approximately 50% of challenged animals developed clinical signs. The total clinical score of the LSD clinically diseased buffaloes reached 9–10, while the score of other animals did not surpass 5, which is a clinical evolution comparable to the patterns previously described [31,32]. Furthermore, the incubation period was different in these two species, with approximately14 days in buffaloes and only 4 days in cattle.

The diagnosis of LSD in the field is mainly based on clinical surveillance, confirmed by laboratory tests performed on blood samples or skin biopsies [33,34]. However, identification of infected subclinical animals remains difficult because nodules are absent and viremia is short or intermittent, making it hard to detect the virus in blood samples [31]. This is particularly true in buffaloes, which develop LSD mainly in the subclinical form [33], with mild or absent clinical signs as also observed in our trial.

One of the most interesting outcomes of the challenge is the lack of LSDV detection in any sample collected from the infected buffaloes, regardless of the development of clinical signs. The presence of the virus in nasal and ocular discharges in infected animals and the amount of virus they shed are crucial to its spread. Clinical signs and molecular results obtained in this trial suggest that buffaloes did not shed the virus, leading to the assumption that they do not have a prominent role in the spread of the disease. These data agree with what was recently reported in a work by Elhaig et al. [33], during Egyptian outbreaks between 2016 and 2018, in which all collected blood and biopsy samples were negative in real-time PCR, but the presence of antibodies confirmed LSDV infection in the animals. Such findings strengthen the hypothesis of buffalo resistance to LSD and its role as an accidental non-adapted host [27,28]. The fact that LSDV DNA was detected neither in blood nor in skin nodules collected from both buffaloes with generalized lesions and those with skin reactions in the inoculation sites, suggests the development of a weak viremia and a scarce distribution of the virus in the peripheral sites once inoculated. As it is difficult to say whether the virus is promptly cleared from the blood circulation and/or not capable of replicating efficiently to evoke the severe generalized form of the disease, further studies are needed to investigate LDSV pathogenicity in buffaloes.

The limited susceptibility of water buffalo against LSD was also confirmed by the different evolution of the infection in the challenged vs. the positive control group. In the challenged group, only two buffaloes showed mild clinical signs (fever, skin nodules) that totally regressed at the end of the trial, while the same inoculum evoked a severe disease in cattle, leading to the euthanasia of cattle n.10 to avoid the suffering of the animal. Both cattle developed a detectable viremia, shed LSDV through nasal and ocular discharge, and showed a peripheral distribution of the virus in the skin nodules detected by qPCR. These results also proved the efficacy of the experimental infection procedure.

Seroconversion was detected in three out of six challenged buffaloes between 39 and 42 days p.i. only by the ELISA test. A recent experimental study conducted on bulls observed seroconversion after 42 days p.i. [10], confirming what we observed in our trial. The short duration of the trial and the late development of the antibody response have probably prevented the record of the seroconversion of the remaining animals. However, the immune response to LSDV infection is predominantly cell mediated [31] and the scarcity of humoral immune response in buffaloes following capripoxvirus infection has been also described in field conditions [35]. In fact, one of the limitations of this study is the lack of any investigation on the cell-mediated immune response. Nevertheless, the ELISA test confirmed its sensitivity in the early detection of anti-capripoxvirus antibodies [7,9,13] in buffaloes compared to the neutralization assay. Indeed, the virus neutralization test is the most specific serological method, but, in this case, lacks sensitivity [36,37]. None of our buffaloes developed neutralizing antibodies during the experimental trial. However, it is hard to say if the lack of circulating neutralizing antibodies is attributable to the short duration of the trial or the immunological response of buffaloes to LSDV infection. However, the poor neutralizing response to LSDV in this species is not a novelty as it had been already reported by several authors. Elhaig et al. [33] did not detect neutralizing antibodies in naturally infected buffaloes with clinical LSD lesions. In a previous work on buffalo sera collected during an interepidemic period, results obtained by ELISA were only partially confirmed by VNT where low neutralizing titers were limited to two buffaloes (1:20) [21].

A different scenario occurred for the surviving cattle, with the detection of neutralizing antibodies from day 14 p.i. followed by positive ELISA results on day 31 p.i., seventeen days later, according to the available literature, highlighting its low sensitivity. We should probably consider that the sensitivity of ELISA and VNT may differ between animal species, as already reported [7].

Whether water buffalo susceptibility to LSD infection could be ascribed to specific genetic variants [38], as for tuberculosis, mastitis, or foot-and-mouth disease [39,40,41,42], is difficult to say but the hypothesis deserves to be further explained in future studies.

## 5. Conclusions

LSD is a cross-border disease, characterized by severe economic losses, that continues to spread worldwide. Real-time PCR is the method of choice for a rapid routine diagnosis in cases of suspicion; however, in species not particularly susceptible to infection such as buffalo, it may also be useful to combine genome detection with serological tests to detect any previous contact with the virus. The present study addresses the hypothesis that water buffaloes are less susceptible to LSDV infection under experimental conditions compared to cattle and that diseased buffaloes may develop mild clinical signs of the disease.

## Figures and Tables

**Figure 1 viruses-16-00466-f001:**
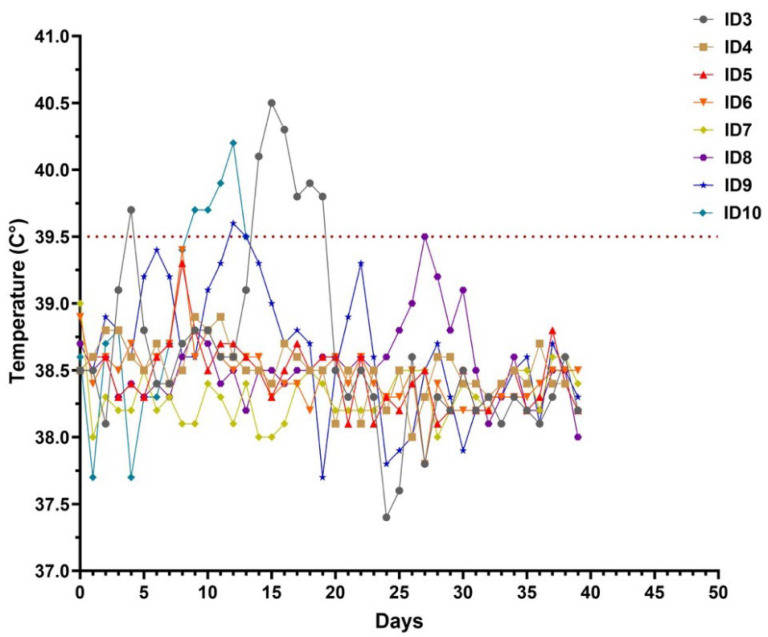
Body temperature during the trial of the six buffaloes and two calves. Buffaloes were identified from ID3 to ID8, while cattle were identified with ID9 and ID10. In cattle ID10, fever had a biphasic course: on day 4, it was 39.7 °C, it rose to 40.2 on day 7, then decreased to 39.5 on day 8, and finally increased again from day 9 to day 10, when it was 40.4 °C.

**Figure 2 viruses-16-00466-f002:**
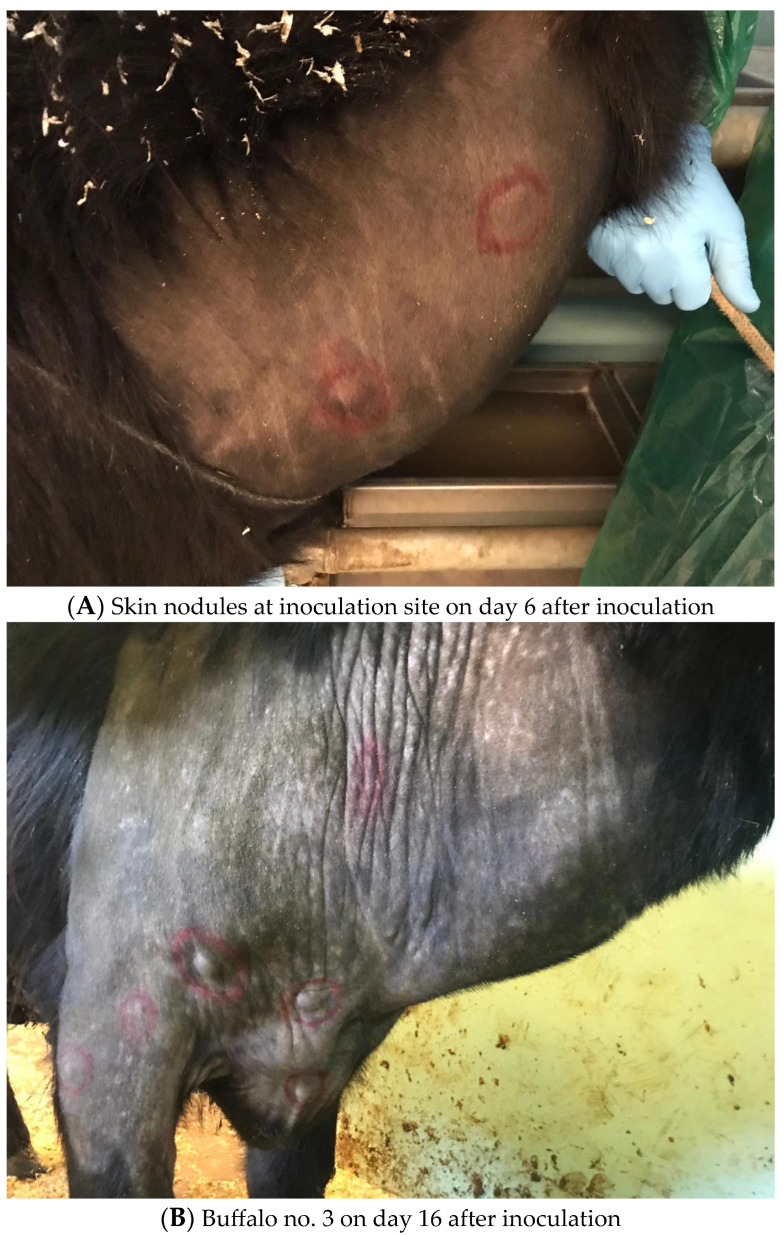
Clinical signs.

**Figure 3 viruses-16-00466-f003:**
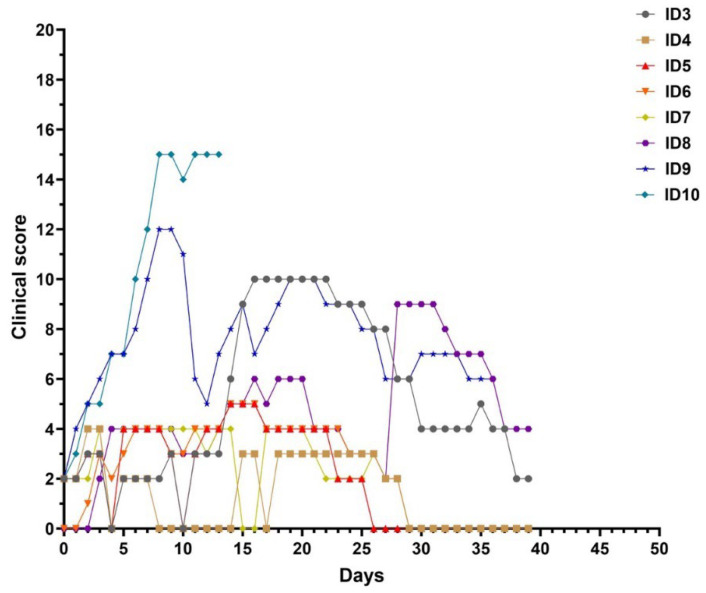
Clinical scores of six water buffaloes and two calves infected with LSDV strain. Buffaloes are identified from ID3 to ID8, whileID9 and ID10 identify cattle. Buffaloes naïve to the virus were infected on Day 0, and clinical scores were assessed every 3 days.

**Figure 4 viruses-16-00466-f004:**
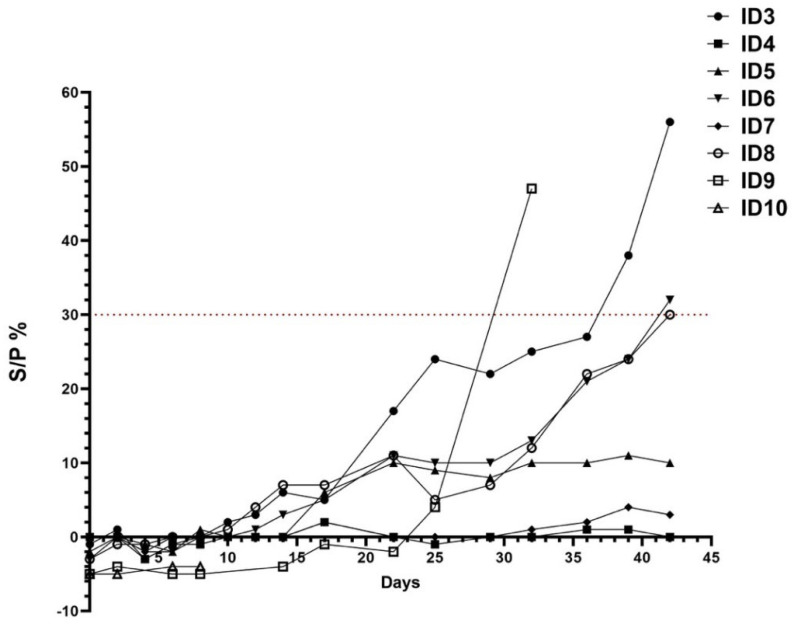
ELISA results (S/P %) over a period of 42 days of buffaloes and cattle. Buffaloes are identified as ID3 to ID8, while ID9 and ID10 identify cattle.

**Table 1 viruses-16-00466-t001:** Clinical scoring system.

General Health Status		Food Intake		Nasal Discharge		Number of Nodules		Dissemination of Nodules		Lymphadenomegaly	
Normal	0	Normal	0	Normal	0	No Nodules	0	No Nodules	0	No Lymphadenomegaly	0
Mild Illness	1	Slightly Decreased	1	Mild	1	≤10	1	Localized	1	Localized	1
		Decreased	2	Marked Mucous	2	<20	2				
Severe Illness	2	Does Not eat	3	Purulent	3	≥20	3	Generalized	2	Generalized	2

The animals were clinically evaluated daily and classified using the scoring system.

**Table 2 viruses-16-00466-t002:** Clinical findings and viremia in six water buffaloes and two cattle used as the positive control group, after experimental inoculation with a field strain of LSDV.

Clinical Finding	Inoculated Animals							
	Buffalo						Cattle	
	3	4	5	6	7	8	9	10 *
Decreased food intake	For 2 days	-	-	-	-	For 2 days	For 2 days	For 4 days
Nodules								
Number	>20	2	4	4	3	12	4	>20
Size (ø)	0.5–5.0 cm	0.2–0.5 cm	0.5–2.0 cm	0.5–2.5 cm	0.2–2.0 cm	0.5–3.5 cm	2.0–6.5 cm	2.0–7.0 cm
Location	Generalized	Inoculation sites	Inoculation sites	Inoculation sites	Inoculation sites	Generalized	Inoculation sites	Generalized
Lymphadenomegaly	Generalized	Prescapular and prefemoral	Prescapular and prefemoral	Prescapular and prefemoral	Not detected	Generalized	Generalized	Generalized
Oedema	ND	ND	ND	ND	ND	ND	ND	Dewlap
Nasal and ocular discharge	ND	ND	ND	ND	ND	ND	From mucopurulent to serous; bilateral	From mucopurulent to serous; bilateral
Viremia	ND	ND	ND	ND	ND	ND	From day 9 (Ct 28.54) to day 14 (Ct 38.40)	From day 5 (Ct 35.57) to day 9 (Ct 32.97) *

* Cattle euthanized on day 10 p.i. ND: not determined.

**Table 3 viruses-16-00466-t003:** Detection of LSDV by real-time PCR in buffalo and cattle tissues and swabs.

	Inoculated Animals							
	Buffalo						Cattle	
Tissue	3	4	5	6	7	8	9	10
Skin	ND	ND	ND	ND	ND	ND	ND	Ct 19.63
Skin nodule	ND ^b^	^a^	^a^	ND	^a^	ND ^b^	^a^	Ct 20.39 Ct 19.39 and 18.47 ^c^
Spleen	ND	ND	ND	ND	ND	ND	ND	Ct 29.14
Kidney	ND	ND	ND	ND	ND	ND	ND	Ct 29.84
Lung	ND	ND	ND	ND	ND	ND	ND	Ct 29.02
Liver	ND	ND	ND	ND	ND	ND	ND	Ct 30.49
Bronchial LN	ND	ND	ND	ND	ND	ND	ND	Ct 29.58
Inguinal LN	ND	ND	ND	ND	ND	ND	ND	Ct 24.11
Mesenteric LN	ND	ND	ND	ND	ND	ND	ND	Ct 28.44
Right prescapolar LN	ND	ND	ND	ND	ND	ND	Ct 35.11	Ct 24.70
Left prescapolar LN	^a^	^a^	^a^	^a^	^a^	^a^	ND	Ct 25.70
Submandibolar LN	ND	ND	ND	ND	ND	ND	ND	Ct 23.63
Rumen	ND	ND	ND	ND	ND	ND	ND	Ct 28.43
Nasal mucosa	ND	ND	ND	ND	ND	ND	ND	Ct 18.73 ^d^

LN, lymph node; ND, not detected in real-time PCR; ^a^, no sample was taken; ^b^, the animal had clinical signs of LSD and the sampled skin nodule was negative real-time PCR; ^c^, nodule located on the back and the scrotum, respectively; ^d^, nodule located on the nasal mucosa.

**Table 4 viruses-16-00466-t004:** Detection of LSDV in swab samples by real-time PCR in experimental infected buffaloes and cattle.

		Day Post Inoculation												
Sample	Animal	2	3	4	5	7	9	14	17	21	24	31	35	38
Nasal swab	Buffalo 3	ND	ND	ND	ND	ND	ND	ND	ND	ND	ND	ND	ND	ND
	Buffalo 4	ND	ND	ND	ND	ND	ND	ND	ND	ND	ND	ND	ND	ND
	Buffalo 5	ND	ND	ND	ND	ND	ND	ND	ND	ND	ND	ND	ND	ND
	Buffalo 6	ND	ND	ND	ND	ND	ND	ND	ND	ND	ND	ND	ND	ND
	Buffalo 7	ND	ND	ND	ND	ND	ND	ND	ND	ND	ND	ND	ND	ND
	Buffalo 8	ND	ND	ND	ND	ND	ND	ND	ND	ND	ND	ND	ND	ND
	Cattle 9	ND	ND	ND	ND	ND	ND	ND	37.12	37.42	39.44	35.92	37.16	ND
	Cattle 10	ND	ND	ND	ND	34.04	31.97							
Oral swab	Buffalo 3	ND	ND	ND	ND	ND	ND	ND	ND	ND	ND	ND	ND	ND
	Buffalo 4	ND	ND	ND	ND	ND	ND	ND	ND	ND	ND	ND	ND	ND
	Buffalo 5	ND	ND	ND	ND	ND	ND	ND	ND	ND	ND	ND	ND	ND
	Buffalo 6	ND	ND	ND	ND	ND	ND	ND	ND	ND	ND	ND	ND	ND
	Buffalo 7	ND	ND	ND	ND	ND	ND	ND	ND	ND	ND	ND	ND	ND
	Buffalo 8	ND	ND	ND	ND	ND	ND	ND	ND	ND	ND	ND	ND	ND
	Cattle 9	ND	ND	ND	ND	ND	37.45	ND	38.45	36.83	ND	34.78	ND	ND
	Cattle 10	ND	ND	ND	ND	37.37	37.97							
Conjunctival swab	Buffalo 3	ND	ND	ND	ND	ND	ND	ND	ND	ND	ND	ND	ND	ND
	Buffalo 4	ND	ND	ND	ND	ND	ND	ND	ND	ND	ND	ND	ND	ND
	Buffalo 5	ND	ND	ND	ND	ND	ND	ND	ND	ND	ND	ND	ND	ND
	Buffalo 6	ND	ND	ND	ND	ND	ND	ND	ND	ND	ND	ND	ND	ND
	Buffalo 7	ND	ND	ND	ND	ND	ND	ND	ND	ND	ND	ND	ND	ND
	Buffalo 8	ND	ND	ND	ND	ND	ND	ND	ND	ND	ND	ND	ND	ND
	Cattle 9	ND	ND	ND	ND	ND	ND	ND	ND	36.93	ND	ND	38.45	ND
	Cattle 10	ND	ND	ND	ND	ND	37.05							

Real-time PCR Ct values are reported for each sample; ND indicates the absence of detectable DNA.

## Data Availability

The raw data supporting the conclusions of this article will be made available by the authors on request.

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
