# Peer review of "Susceptibility of Mediterranean Buffalo (Bubalus bubalis) following Experimental Infection with Lumpy Skin Disease Virus"

_viruses, 2024, doi:10.3390/v16030466_

Round 1

Reviewer 1 Report

Comments and Suggestions for Authors

Comments on the Quality of English Language

Author Response

1. The title was rewritten in line with rules of English language as suggested.

2. Abstract – this section was improved by reorganizing results and by adding relevant conclusion.

3. Virus origin, isolation, stock production and quality control . The field virus used for challenge test was isolated from a symptomatic calf skin nodule. The sample was collected during an outbreak occurred in Albania in 2017 and was kindly provided by Dr. Ledi Pite and Dr. Liljana Cara working at the Food Safety and Veterinary Institute (FSVI) in Tirana. Briefly, the biopsy was weighed, ground in a mortar, and suspended 1:10 (w/v) inphosphate-buffered saline (PBS) pH 7.4  plus 10,000 IU/mL penicillin (Sigma), 5,000 IU/mL nystatin, 10 g/L streptomycin sulfate salt (S9137, Sigma), and 125 mg/L gentamycin sulfate (G1397, Sigma). The suspension was frozen at 80°C degrees and thawed three times, centrifuged at3000xg for 30 min at 4 °C, and the supernatant used for virus isolation. Virus stock was obtained by five consecutive passages on MDBK cells (ATCC -CCL-22) a continuous cell line susceptible to LSD virus infection..Cells were incubated at 37 °C in an atmosphere of 5 % CO2 and observed daily using an inverted microscope (20-40X Leica DFC425 C, Leica Microsystem Ltd.) to evaluate the presence of virus–specific cytopathic effect (CPE).When the CPE was complete cells were frozen at -80°C degrees and thawed three times and the suspension harvested. After centrifugation at 3000 xg for 30 min at 4 °C,the supernatant was collected, distributed into aliquots and stored at -80 °C. One aliquot of supernatant was verified for sterility for bacteria, fungi and mycoplasma as prescribed in the European Pharmacopoeia. Another aliquot was used to determine the viral titer, for virus identification in qPCR, and to evaluate its purity (absence of contaminant viruses) as prescribed in the European Pharmacopoeia.

Line 186: cell concentration in VNT Madin-Darby bovine kidney MDBK cells (ATCC –CCL-22) used at passages from 150 to 160 and prepared at concentration of 10 5 cell/ml; 100 µl added to each well on 96-well plate to obtain a final concentration of 10 4 cell/well in VNT.

LSDV Neethling attenuated vaccine strain used for VNT was obtained from OBP (Onderstepoort Biological Products) freeze-dried live attenuated virus preparation, the commercial product for the prophylactic immunization of cattle against Lumpy Skin Disease. The virus was amplified using confluent monolayer of MDBK cells (ATCC -CCL-22) at 37 °C until 90–100 % cytopathic effect (CPE) was observed. Harvested virus was titrated and stored at -80°C until required.

4. a-b) clinical observations (temperature and clinical score) have been presented in graphs using a different program for graphical presentation and figure descriptions have been added below the figures as suggested.

c) Serology was presented in a graph, but it was impossible to do the same for PCR data.

5. Discussion- this section was partially rewritten and reorganized. Limitations of the study were added. At the end of this section (382-384) low sensitivity of ELISA was discussed: “A different scenario occurred for the surviving calf, with the detection of neutralizing antibodies from day 14 p.i. followed by positive ELISA results on day 31 p.i., seventeen days later, according to the available literature, highlighting its low sensitivity. We should probably consider that the sensitivity of the ELISA may differ between animal species, as already reported [7]”. 

Reviewer 2 Report

Comments and Suggestions for Authors

The manuscript titled “Susceptibility of Mediterranean buffalo after experimental infection with lumpy skin disease virus” describes the experimental infection of six Mediterranean buffalo and two cattle with a virulent strain of LSDV, whilst two Mediterranean buffalo were mock inoculated. These ten animals were observed, sampled and subsequent serological and molecular analyses were performed for 42 days. The manuscript is interesting and contributes to the flurry of new studies describing LSD outbreaks in various host species other than cattle. Unfortunately, there are a couple of concerns listed below:

1. The manuscript would benefit from language curation.

2. The inability to detect LSDV in any of the samples is a concern, since it contradicts to the first description of LSD in water buffalo [Ref: 18]. Especially in buffalo No 3 and 8 that showed generalized and well-circumscribed nodules. Was an endogenous internal control used during the qPCR? Was virus isolation attempted from these larger nodules?

3. In the absence of detecting LSD DNA, it would be beneficial to include pictures of the symptoms observed in the buffalo.

4. It would be advantageous to mention the species name: Mediterranean buffalo (Bulbalus bubalis), in the title.

5. Please check the dates of outbreaks in the Introduction (Lines 67 – 74).

Comments on the Quality of English Language

 The manuscript would benefit from language editing.

Author Response

1. The manuscript was revised in line with rules of English language

2. No internal control was used for the qPCR, as described by Bowden et al., 2008. We attempted to isolate the virus in different cell cultures (OA3.Ts cells and MDBK cells) from one skin biopsy, but no virus was isolated.

3. Pictures of buffaloes with skin nodules were added as suggested by reviewer

4. Species name, Bubalus bubalis, was added to title

5. Lines 67-74 were taken from EFSA _Assessment of Control Measure for Category A Diseases of Animal Health Law: Lumpy Skin Disease.

Round 2

Reviewer 1 Report

Comments and Suggestions for Authors

I have checked and all my comments were addressed.  

I suggest to accept the manuscript.

Reviewer 2 Report

Comments and Suggestions for Authors

The manuscript titled “Susceptibility of Mediterranean buffalo after experimental infection with lumpy skin disease virus”, has improved significantly with the addition of the disease photos. Yet, the manuscript can still be improved by additional language curation. 

Line 77-78: The first reports of LSD in Russia was 2015, not 2020 as indicated.

Line 145: …and virus identification using real time PCR (not qPCR)?

Currently two figure 2 in the manuscript.

Comments on the Quality of English Language

The manuscript can still be improved by additional language curation. For example:

Line 25: …”anti LSDV antibodies were”

Line 41: Clinical infection range from mild sub-clinical to acute and …..

Line 44: ….”resulting in a decrease in milk”…..

Line 71: In 2015, the first report of LSD in Europe came from Greece with more than 100 outbreaks reported.

Line 87: …in the last ten years its population increased by 50%.

Line 197: Serological analysis was performed using the commercial…

Author Response

The  manuscript was improved by additional language curation as suggested by reviewer.

Lines 77-78: the date was change as suggested by the reviewer : "LSD outbreaks were reported in Armenia and Russia in 2015"

Line 145: real time PCR, no a quantitative PCR.

The numbering of the figures has been corrected